# Superantigens, a Paradox of the Immune Response

**DOI:** 10.3390/toxins14110800

**Published:** 2022-11-18

**Authors:** Sofia Noli Truant, Daniela María Redolfi, María Belén Sarratea, Emilio Luis Malchiodi, Marisa Mariel Fernández

**Affiliations:** Cátedra de Inmunología—IDEHU (UBA-CONICET), Facultad de Farmacia y Bioquímica, Universidad de Buenos Aires, Junín 956 4 Piso, Ciudad Autónoma de Buenos Aires 1113, Argentina

**Keywords:** staphylococcal superantigen, enterotoxin, toxin pathogenicity, immunomodulation, molecular and cellular targets

## Abstract

Staphylococcal enterotoxins are a wide family of bacterial exotoxins with the capacity to activate as much as 20% of the host T cells, which is why they were called superantigens. Superantigens (SAgs) can cause multiple diseases in humans and cattle, ranging from mild to life-threatening infections. Almost all *S. aureus* isolates encode at least one of these toxins, though there is no complete knowledge about how their production is triggered. One of the main problems with the available evidence for these toxins is that most studies have been conducted with a few superantigens; however, the resulting characteristics are attributed to the whole group. Although these toxins share homology and a two-domain structure organization, the similarity ratio varies from 20 to 89% among different SAgs, implying wide heterogeneity. Furthermore, every attempt to structurally classify these proteins has failed to answer differential biological functionalities. Taking these concerns into account, it might not be appropriate to extrapolate all the information that is currently available to every staphylococcal SAg. Here, we aimed to gather the available information about all staphylococcal SAgs, considering their functions and pathogenicity, their ability to interact with the immune system as well as their capacity to be used as immunotherapeutic agents, resembling the two faces of Dr. Jekyll and Mr. Hyde.

## 1. Introduction

The term superantigen (SAg) was introduced by White et al. (1989) to describe a group of bacterial proteins targeting, as unprocessed molecules, the variable portion of the β chain of the T cell receptor (TCR) and the major histocompatibility complex type II molecules (MHC-II) expressed on antigen-presenting cells (APC). As a result of this simultaneous interaction, there is a massive activation of the immune system along with an intense proliferation of T cells, either CD4+ or CD8+ [1,2]. Although the term superantigen was introduced in 1989, these toxins had been described in the early 1980s as the causative agent of the highly lethal Toxic Shock Syndrome (TSS) associated with tampons [3,4,5], which was later explained by the deregulation of the immune system caused by SAgs characterized by generalized multiple organ failure caused by a pro-inflammatory cytokine storm. SAgs are also responsible for food poisoning and triggering autoimmune diseases in sensitive hosts, among other conditions. They can promote immunosuppression in the infected host, allowing bacterial spread and further sepsis [6,7,8].

SAgs are produced by Gram-positive bacteria such as *S. aureus* and *Streptococcus pyogenes*, but they have also been found in other species such as β-hemolytic streptococci, coagulase-negative staphylococci, Gram-negative *Yersinia pseudotuberculosis*, *Pseudomonas fluorescens* and cell wall-less bacteria *Mycoplasma arthritidis* [9,10,11,12]. Furthermore, SAgs encoded by murine and human retroviruses have also been described [13,14,15].

Until now, the best characterized SAgs were produced by *S. aureus* and *S. pyogenes*. Although these two bacteria are not phylogenetically related species, staphylococcal and streptococcal SAgs share sequence homology and biological similarities. Some SAgs from these two species are so close that they belong to the same group or family, as is very well described for the SE family or Group II, where the streptococcal superantigen SpeA is more similar to staphylococcal enterotoxin B (SEB) than to staphylococcal enterotoxin G (SEG). SpeA displays a low degree of similarity with the other streptococcal SAgs. This finding supports the horizontal transmission of genes between species. This theory is also reinforced by the encoding of *sags* in genetic mobile elements, such as plasmids or transposons [16].

Interestingly, the intense immune stimulation induced by SAgs could allow the spread and further colonization of the infected host by the pathogen instead of favoring its eradication. This characteristic would turn superantigen activity into a paradox of the immune response.

## 2. Staphylococcal SAgs

### 2.1. An Overall Description

*S. aureus* is one of the major pathogens responsible for human and veterinarian diseases causing mild to life-threatening infections. Human beings are the primary reservoir for this bacterium. It is well known that about 20–30% of the population persistently carries this microorganism in the anterior nares, while about 50–60% are intermittently colonized [17,18,19,20]. The colonization of other extra nasal sites, such as the skin and gastrointestinal tract [21,22,23,24] has also been reported. It should be mentioned that colonization is considered one of the main risk factors for *S. aureus* diseases [25].

This pathogen harbors several virulence factors among which some are surface proteins and many others are secreted. Staphylococcal enterotoxins (SEs) or SAgs are some of the most important and are associated with both colonization and pathogenicity of *S. aureus*.

The first staphylococcal SAgs described were the classical SAgs, including staphylococcal enterotoxins A to E and the non-emetic toxin TSST-1. Due to its strong association with TSS, the toxin formerly called SEF was later named Toxic Shock Syndrome Toxin number 1 or TSST-1 [26,27,28,29,30,31,32,33]. Since that first group was described, 24 more SAgs were identified, and were considered non-classical or new SAgs.

The first non-classical SAg, SEH, was described in 1994 [34]. Later, SE*l*J was acknowledged [35] at the same time that Munson et al. described two new SEs, G and I [36]. In 2001, Jarraud et al. defined the *egc* operon for the first time, as an enterotoxin gene nursery encoding for the proteins SEO, SEM, SEI, SEN and SEG (firstly named SEL, SEM, SEI, SEK and SEG) and two pseudogenes, ψ ent1 and ψ ent2 [37]. Moreover, SEK and SEL were simultaneously described [38,39,40]. In the early 2000s, genetic variations within the *egc* cluster were described, which allowed the identification of SElU and SElV [41,42].

Later on, SEP [43], SES and SET [44], SEQ [45], SER [46], SE*l*W [47], SE*l*X [48] SE*l*Y [49] and SE*l*Z [50] were described. Lastly, SE01 [51], SE02 [52], and SE*l*26 and SE*l*27 were identified and characterized among the complex form by *S. aureus*, *S. argenteus* and *S. schweitzeri* [53]. Table 1 summarizes the information described above. Phylogenetic group correspondence is explained below in Section 2.4.

According to the International Nomenclature for Staphylococcal Superantigens, *S. aureus* enterotoxins (SEs) are defined by their emetic ability when ingested, while the term SAg considers the effects on the immune system. Taking that into account, only the toxins that induce emesis after oral administration in a primate model are designated as SEs. Other related toxins that either lack emetic properties in this model or have not been tested are defined as “staphylococcal enterotoxin–like” (SE*l*) SAgs, to indicate that their potential role in staphylococcal food poisoning has not been confirmed [54].

To date, TSST-1 and SE*l*J are the only tested superantigens with non-emetic properties, along with a report of a truncated SE*l*X [55].

Furthermore, some of the so-called SE*l* and SE proteins have already been tested for their emetic properties in a proposed model using house musk shrews *(suncus murinus*) [56,57]. In this model, in addition to all the other SEs, SE*l*Y has been tested and shown to have emetic properties [44,49,56,58,59]. Recently, a new emetic animal model was established using common marmosets and, in this case, SEA, SEB, SEC, SEI, SE*l*Y and SE02 showed emesis-inducing activity, while TSST-1 did not [52,60].

It has been shown that almost every isolated strain of *S. aureus* carries at least one *sag* gene in its genome [61,62,63,64]. Although there are some differences in the percentages of total gene prevalence reported, a lower prevalence of *sag* genes (70–85%) was found when a few *sag* genes were analyzed [61,65], whereas in other studies evaluating a larger number of genes, the percentages were almost 98% [66,67]. Noteworthy, as mentioned before, 30 SAgs have been hitherto described, which suggests some underestimation of SAg prevalence. Additionally, more than 50% of the isolates encode for the *egc* operon—with eight different forms described—being that the *egc* SAgs are the most prevalent genes [37,62,65,68,69,70,71]. However, there are some differences in the methodology used to define the presence of the operon; while some papers considered all *egc* SAgs, others picked two (such as *seg* and *sei* or *sem* and *seo*) without considering the operon variants, which in some cases may underestimate the prevalence of the operon itself. With regard to this issue, a consensus method should be established to assess the presence of the *egc* operon, which should definitely consider all the toxins involved in the operon, taking into account possible DNA mutations in their genes.

### 2.2. Production and Detection of Staphylococcal SAgs

Although there are some statements about the production of almost every SAg, in addition to the 6 classical ones (SEA-SEE and TSST-1), there is not much information regarding the individual gene expression and the mechanism of regulation involved in their production. However, many factors and pathways have been described to impact positively and negatively in the production of SAgs and other virulence factors [72,73,74].

Considering the classical SAgs, early reports using mainly single or double-gel diffusion indicated that SEB and SEA are produced in all the growing phases [75,76,77,78]. In contrast, SEC seems to be produced during the exponential growth phase or at the early stationary phase [79,80]. In regard to SED, mRNA expression appears to be higher in the stationary growth phase using qPCR [81], and evidence of its production with SEE at the stationary phase has also been provided [82].

With respect to TSST-1, although its production occurs principally at the stationary phase, it was found that several environmental elements are capable of triggering or inhibiting its production, including temperature, carbon dioxide atmosphere and antibiotic pressure [16,83,84,85].

Many authors rely on the conclusions reported by Derzelle and collaborators to describe the patterns of staphylococcal enterotoxins expression [86]. These authors found a differential expression behavior between classical and new SAgs. Their work is limited to mRNA, as they performed RT-qPCR assays; however, protein secretion was not evaluated. Furthermore, although their analysis was performed in a total of 28 strains, the number of strains used per gene was variable and dependable on the sag gene profile of each strain, between one and thirteen. In the case of the *egc* operon, only *seg* and *seu* were evaluated in all *egc* positive strains (thirteen), and the remaining genes (*sei*, *sen*, *sem* and *seo*) were only evaluated in three strains. Therefore, these results should be validated using other strategies for reliable conclusions. A similar analysis for SE02 showed the production of this SAg in vitro in the early exponential phase and to a lesser extent in the stationary phase, with these results confirmed by Western blotting [52].

In vitro assays indicate that *egc* superantigens are produced at much lower concentrations than classical SAgs TSST-1, SEB and SEC, which may explain the relevance of these toxins in pathogenesis [87,88,89]. Particularly, it was proved that TSST-1 was produced in a high level in biofilms, which could be transferred to production on mucosal and skin surfaces [89].

Some evidence suggests that the regulation of gene expression is not the same for *S. aureus* in vitro than during infection in vivo [90,91], showing differences in SAg expression even between tissues or isolation sources [82,92]. As a result, there is a different SAg expression despite the evaluation of DNAs encoding the same *sag* genes. In addition, it has been proposed that the detection of SAg mRNA does not correlate with the presence of toxins [93].

Proteomics approaches confirmed the production of some toxins such as SEB, SEC, SEH, SEK and SEQ both in exponential and stationary phases, and TTST-1, SEL, SE*l*U and SEP in the stationary phase [84,94,95,96], but the whole exo-proteome was analyzed in vitro; therefore, relevance in vivo has not been clarified [97]. In addition, a wide heterogeneity between the proteins identified amongst the different isolates analyzed in vitro [98,99] has been reported.

Although PCR is the chosen method to identify the presence of *sag* genes in bacterial DNA, it is not effective to evaluate the presence of SAgs in patients, food or in vitro-generated samples [100,101,102].

At present, ELISA-based methods are considered the best option for the detection and quantification of SEs/SE*l*s in several samples because of their robust and sensitive results, with reported detection limits between 0.25 and 1 ng of SAgs per g of sample [102]. However, there are no available ELISA kits for each of the 30 SAgs identified, which complicates the evaluation of the full profile of SAgs potentially produced by *S. aureus*. An approach based on a combination of chromatography and MS measures simultaneously a group of toxins; however, it is not easy to conduct and does not cover all existing SAgs [103].

In addition, Surface Plasmon Resonance (SPR) was also postulated as a method to detect SAgs in different biological samples, showing a limit of detection in the picomolar range, with the advantage to measure simultaneously three or more SAgs present in the same assay depending on the biosensor used [104,105,106,107,108,109].

All in all, each method has its advantages and disadvantages and there is no ideal technique to assess the production of all toxins; however, it is important to bear in mind the limitations of the method used at the time of reaching conclusions.

### 2.3. Superantigens and Human Diseases

#### 2.3.1. Toxic Shock Syndrome

Superantigens have been associated with many illnesses caused by *S. aureus* infections. One of the most well-recognized diseases is Toxic Shock Syndrome (TSS), a potentially lethal febrile illness related to multiorgan dysfunction that occurs as a consequence of the cytokine storm produced by SAgs. TTS is characterized by hypotension; diarrhea; labored breathing; and changes in heart, liver and kidney function [110,111,112]. This pathology has a high incidence in women during their menstrual periods, and is particularly called menstrual TSS (mTSS) [112]. TSST-1 is associated with all menstrual cases, in some instances, coproduced with SEA or SEC-1; conversely, SEB production has been negatively related to this syndrome [111,113,114,115,116]. Regarding non-classical SAgs, Jarraud and collaborators suggested that the production of SEG and SEI by *S. aureus* strains that do not produce TSST-1, SEA to SEE and SEH and isolated from mTSS patients may explain these clinical manifestations [117].

Studies in porcine vaginal tissue have proved that, initially, the presence of *S. aureus* triggers the innate immune system activation and increases TSST-1 flux across the mucosa [118]. Furthermore, it has been demonstrated that TSST-1 is essential to stimulate systemic inflammation by inducing the production of IFN-γ and suppressing autophagy [119,120].

TSST-1 biochemical and biological properties have been studied, and among them, its capacity to interact with CD40 on epithelial cells. This interaction promotes chemokine production and attracts other components of the adaptive immune response. Consequently, local inflammation occurs and provokes mucosal disruption, facilitating TSST-1 penetration and enhancing mTSS [121].

Moreover, mTSS progress has been associated with oxygen introduction into the anaerobic vagina. This phenomenon was related to the use of medical devices to control menstrual flow, especially certain tampons whose composition enhanced their ability to trap oxygen within its fibers. Remarkably, device insertion does not cause significant vaginal oxygenation. In addition to the introduction of oxygen, the increase in body temperature, as well as protein, low glucose levels and neutral pH generate an environment that is favorable for TSST-1 production. In addition, the device’s capability to increase the permeability of the mucosa promotes SAg penetration [114,122,123,124,125].

Moreover, it has been reported that tampon wearing time influences the development of mTSS [122]. Some studies propose that the tampon material also affects *S. aureus* proliferation and TSST-1 production, while others suggest that the tampon structure and fiber density have a higher impact on colonization and toxin production than the nature of its material [124,126]. It has been described that the application of menstrual cups could promote *S. aureus* growth and biofilm formation with toxin production. Although those conditions are affected by cup composition, toxin production induced by cup use is lower than that caused by tampon use. This fact has to be related to its incapacity to trap oxygen [124,125].

Non-menstrual TSS has also been associated with TSST-1 and other classical superantigens such as SEB, SEC1 and SEA [61,113,114,115,127,128,129]. A few studies suggest that SEB production is prevalent in *S. aureus* isolates that are negative for TSST-1 in non-menstrual TSS cases [113,129]. Furthermore, in case-control investigations, it was demonstrated that *sea* gene is correlated with TSS, whereas the presence of *sem* and *seo* genes is correlated with sepsis [61].

#### 2.3.2. Infection Endocarditis

Infection endocarditis (IE) is a disease of the endocardium and cardiac valves, that causes cardiac and multi-organic symptoms usually as a consequence of *S. aureus* bacteremia [130,131]. Classical superantigens, such as TSST-1, SEC, SEB, SEA, SED and SEE are related to IE, as well as non-classical, from the *egc* operon [87,132,133,134,135]. It has been shown that TSST-1 and *egc* operon gene deletion raised mortality in a rabbit model. Many studies have associated disease development with vegetation formation in aortic valves, whose formation decreases with the deletion of *egc*, SEC and TSST-1 genes [87,133]. 

Another action mechanism that could influence the inflammation process is the secretion of IL-8 by aortic endothelial cells induced by SEC [134]. In addition, it has been demonstrated that SEC inhibits the pro-angiogenic factor serpin E1 impeding reendothelialization and promoting disease [133]. On the other hand, TSST-1 also acts on the endothelium, producing upregulation of vascular and intercellular adhesion molecules (VCAM-1 and ICAM-1), raising permeability and affecting tissue re-composition [135].

#### 2.3.3. Pneumonia

Pneumonia is an infectious respiratory disease frequently related to *S. aureus* infections in both community and hospital settings [136]. Clinical features include cough, fever, tachypnea, tachycardia and pulmonary crackles, leading to severe pulmonary infections [137,138]. Staphylococcal pneumonia caused by methicillin-resistant Staphylococcus aureus (MRSA) has been of growing concern due to its resistance to several antibiotics and its higher risk of morbidity and mortality [139,140]. It has been observed that several isolates obtained from patients with lung damage produced TSST-1, SEB and SEC in vitro [141,142]. Spauding et al. [33] proposed that SAgs contribute to pneumonia development once they are produced by *S. aureus* in the respiratory tract, generating an intrapulmonary inflammation. In addition, the cytotoxic effect on endothelial cells of the alveolar–capillary barrier may cause edema and respiratory distress. Furthermore, T cell activation and the release of cytokines may contribute to the pathogenesis and would be associated with the clinical symptoms [143,144]. However, there is no solid evidence proving that SAgs are the cause of pneumonia in humans.

#### 2.3.4. Staphylococcal Food Poisoning

Staphylococcal food poisoning (SFP) is a foodborne disease that provokes abdominal pain, intense diarrhea and vomiting [2]. This pathology has been related to many SAgs in investigations from all over the world. The presence of classical SAg genes, such as *sea*, *seb*, *sec*, *sed* and *see*, in *S. aureus* isolates from dairy products, animals and infected patients has been associated with SFP development [101,145,146,147,148,149,150]. In those samples, not only classical SAg genes, but also *egc* genes such as *sei*, *seg*, *sem*, *sen*, *seo* and *seu* have been detected [101,145,147,148,150,151,152,153]. Moreover, other non-classical SAg genes, such as *seh*, *sej*, *sek*, *sep*, *seq* and *ser*, have been identified in isolations related to SFP [101,145,147,149,150,154,155,156,157].

While genetic studies and genomic patterns are relevant to the study of this illness, the confirmation of the physical presence of toxins in food products suspected of contamination or an analysis of isolation capability of expressing SEs is needed to verify their contribution to SFP. Notably, some isolations from SFP patients and milk samples containing *seg* and *sei* genes did not produce detectable levels of SEG and SEI. On the other hand, isolations harboring *seh* were able to produce significant levels of this SAg. Moreover, some positive isolations for *sea*, *seb* and *sec* did not express those toxins in vitro [93,158]. Other studies showed the presence of SEC, SED, SEA, SEB and SEE in samples related to SFP cases and their expression in isolates from sick people or food [159]. SEH and SEA expressions were also shown in isolates from foodstuffs causing poisoning events [154,155]. SEI and SEM production in isolates associated with food poisoning was also corroborated in several cases [160].

A variable distribution of *S. aureus* genotypes and SEs expression has been shown worldwide, suggesting that the distribution of SAg genotypes may be related to geographical distribution [156,161].

Some mechanisms of action for vomiting and diarrhea have been proposed in several animal models. Many studies suggested that enterotoxins stimulate the vagus nerve in the gut, which transmits the signal to the vomiting center in the brain [162,163]. Supporting this idea, a few studies have shown that SEA increases 5-hydroxytryptamine (5-HT) and histamine release in the intestinal tract, which would increase the discharge of vagal afferent fibers and trigger an emetic effect; it was also shown that this SAg needs vagal 5-HT3 receptors to stimulate vomiting [60,162]. Additionally, it has been described that some SAgs, such as SEA, SEB, SEE and TSST-1, can penetrate the gut lining. Interestingly, it is proposed that transcytosis is mediated by an amino acid sequence highly conserved across superantigens [164]. After transcytosis, the activation of the local immune system occurs. Using a mice model, it was shown that SEB administration resulted in an early expansion of peripheral T cells and Th1 cytokine secretion [165], and that CD4+ T cell stimulation by SEB provoked a disruption of the jejune tonic and stimulated ion transport, creating water movement and contributing to a diarrheal response [166]. Furthermore, SEA and SEB could interact with the MH C II of human subepithelial intestinal myofibroblasts, triggering the secretion of pro-inflammatory cytokines. These effects may play a role in inflammatory injury [167].

#### 2.3.5. Autoimmune Pathologies

SAgs have been correlated with the onset of autoimmune pathologies [168,169]. SAgs may activate autoreactive T and B cells. In addition, APC activation may lead to alterations in antigen processing, with the production and presentation of autoantigens as a consequence [170].

##### Kawasaki Disease

Kawasaki disease (KD) is an autoimmune, febrile and inflammatory syndrome that involves coronary vasculitis. It is reported that this illness may have different etiologies such as viral and bacterial infections, imbalance in immune response and genetic factors [171,172,173,174].

Although, individuals that develop KD may have a genetic predisposition, some theories propose that SAgs could be implicated in massive immune stimulation, activating T cells, B cells and macrophages, and promoting a considerable pro-inflammatory cytokine secretion, allowing vasculitis development [173,174]. A proposed mechanism of action suggests that superantigens, secreted by *S. aureus* in the gut microbiota, may penetrate the mucosal membrane and activate Th1, T cytotoxic and B cells, generating autoreactive cells capable of enhancing illness development [173,175].

SAgs secretion by *S. aureus* isolations from patients with KD has been demonstrated in several cases [176,177,178]. Moreover, IgM antibodies against TSST-1, SEA, SEB and SEC were detected in KD patients’ sera, and titers were significantly higher than in the control group [179,180].

In contrast, some authors state that there is no evidence of SAg involvement in the pathogenesis of KD, or that it remains unclear [171,181,182].

##### Diabetes Mellitus

Some animal models and in vitro experiments suggest that SAgs could have a role in diabetes mellitus II (DMII) progression. Chronic exposure to sublethal doses of TSST-1 in a rabbit model showed that this SAg induced glucose tolerance in vivo and systemic inflammation. In addition, SAg-treated adipocytes produced a higher level of pro-inflammatory cytokines (IL-6, IL-8 and TNF) in comparison to the control group. Moreover, TSST-1 induced lipolysis and insulin resistance in adipocytes. In addition, the liver damage shown in the TSST-1 group could lead to an increase in circulating endotoxin levels, which could impact the development of insulin resistance in DMII. Altogether, it was postulated that SAgs may promote insulin resistance and systemic inflammation, leading to disease worsening [183].

Other investigations on adipocyte metabolism showed a possible relationship between SAgs and insulin resistance and cytokine production. A study showed that SEA interacts with the human cytokine receptor gp130, which is present ubiquitously in human cells, and particularly in adipocytes, and consequently reduces insulin signaling and modulates glucose uptake. Binding was mediated by a structure that is well-conserved in several other SAgs [184]. In addition, it was demonstrated that *egc* SAgs, SEI, SEG, SEM and SEO could bind gp130, by SPR [62]. Furthermore, the effect of TSST-1 and SEB in adipocyte cytokine production was analyzed, showing that they provoke IL-6 and IL-8 secretion, which could contribute to inflammation and diabetes persistence [185].

Despite the previous reports, there are no significant studies in humans that demonstrate that there is a significant correlation between DMII development and chronic SAg exposure; thus, further investigations should be conducted.

##### Rheumatoid Arthritis

Rheumatoid arthritis (RA) is an autoimmune disease characterized by inflammation and infiltration of synovial tissues that can lead to important damage in articulations [186]. It has been suggested that SAgs could trigger the disease by activating autoreactive T cells, could have cross-reactivity with self-antigens and may contribute to long-term inflammation [186,187].

Numerous studies in RA patients demonstrated the presence of SEC, SEA and SED and their genes in blood samples, synovial fluids and DNA extracts from those samples [188,189,190]. The *see* gene was present in several samples of synovial fluid of RA patients, but the expression of the toxin was not analyzed [191]. SEA, SEB, SEC and TSST-1 were also identified in the synovial fluids of a group of pediatric RA patients [192]. All these studies have a limitation in common: they did not assess other SAgs.

In addition, some studies found antibodies against SEB and TSST-1, with significantly higher titers in RA patients than in healthy groups [193,194].

Several animal models were used to investigate the correlation between SAgs and RA development and, as mentioned earlier, it was proposed that the inflammatory response and the reactive cells generated by SAgs play an important role in the development of the illness and its reactivations and severity [195,196,197,198,199]. It was suggested that pro-inflammatory cytokine secretion induced by SAgs may enhance the expression of MHC II in synoviocytes, promoting the release of chemotactic factors and cell infiltration. In addition, SEA and SEB significantly increase the proliferation of T cells in co-culture with synoviocytes from RA patients, compared to the control group. These results suggest that the interaction between both cells and SAgs may be important in disease pathogenicity [200].

##### Atopic Dermatitis

Atopic dermatitis (AD) is a pruritic skin disorder that presents with eczema, inflammation and immune cell infiltration into the local skin lesion [201]. Chronic exposure to SAgs may be a factor that promotes disease incidence, as *S. aureus* colonization is usual in AD patients [202,203]. It is proposed that the disease is caused by inflammatory and allergic factors, and SAgs may be implicated in both. On the one hand, SAgs can activate T cells promoting pro-inflammatory cytokine release and contributing to skin injury, enhancing the Th1 profile. On the other hand, SAgs may activate B cells promoting SAg-specific IgE which can stimulate basophile and mast cell degranulation, leading to a Th2 response [204].

The relationship between SAgs and AD has been based on several factors, such as the presence of anti-SAgs IgE in sera from patients with AD. Anti-SEA, SEB and TSST-1 IgE has been documented in several AD patients with significantly higher titers than in the control groups [201,205,206,207]. Moreover, a total IgE increase has been positively correlated with a SAg-specific IgE increment in patient sera, and it was demonstrated that SAg-specific IgE titers increased with the severity of the condition [206].

In addition, SEA, SEB and TSST-1 provoked a SAg-specific histamine release by basophils from AD patients after being sensitized with IgE-SAg specific serum, in contrast to the control group. Notably, basophil samples from patients lacking these antibodies did not secrete histamine. These results showed that SAgs and their interaction with basophils mediated by IgE could exacerbate AD, promoting an allergic response [201].

Another important fact, which relates SAgs and AD development, is the production of SAgs by *S. aureus* isolated from patients. Compared with control groups, these isolates had a significantly higher production of SAgs and greater heterogeneity of encoded genes. Moreover, they were more likely to produce TSST-1, SEB and SEC [208]. In addition, *S. aureus* isolated from AD patients produced TSST-1, SEA, SEB, SEC and SED [201,209,210].

SAgs are able to induce these pathologies due to their ability to promote a pro-inflammatory environment by upregulating co-stimulatory molecules, which are usually expressed at low levels, preventing the stimulation of autoreactive lymphocytes. Another important point to note is the globular and compact structure shared by these toxins, which is the key to remaining as a whole and active molecule during the passage through the lysosome vesicle and reaching again the dendritic cell surface in the lymph nodes, where they are able to activate different lymphocyte T populations by the crosslinking of the TCRs in association with the MHC-II molecules and other recently described molecular targets, such as CD28 and B7 [211,212,213,214]. In addition, their ability to interact with diverse molecular targets in different types of cells opens up a new avenue to study their functions in new pathologies.

### 2.4. Staphylococcal Classification, Structure and Molecular Targets

Staphylococcal SAgs are homolog proteins that share from 20 to 89% similarity in their amino acid sequence (Appendix A). Based on their sequence of nucleotides and amino acids, SAgs were separated into five evolutive groups or families: Group I, which includes TSST-1, SET, SE*l*X, SE*l*Y and members of the superantigen-like toxins; Group II, including SEB, SECs, SEG, SER, SE*l*U, SE*l*W, SE*l*Z, SEl27 and SE02; Group III, comprising SEA, SED, SEE, SE*l*J, SEH, SEN, SEO, SEP, SES and SE01; Group V, gathering SEI, SEK, SEL, SEQ, SEM SE*l*V and the recently described SE*l*26. Group IV is represented by another group of toxins produced by *S. pyogenes* [33] (Table 1).

Staphylococcal and streptococcal superantigens share a common three-dimensional arrangement comprising an N-terminal β-barrel domain (OB-fold) and a C-terminal β-grasp domain with β-sheets that wrap around a long α-helix. TSST-1 is the only SE described that is less related to the other SEs from a structural point of view. Despite this fact, the basic tertiary structure present in TSST-1 is also found in the other SEs, even though these toxins have more structural complexity. This overall structure is represented in Figure 1. The loop known as the disulfide loop is located between the β4 and β5 strands. Due to its flexibility, no electron density from the crystallographic data is recorded. Experimental data suggest that this loop is responsible for the emetic properties of the SEs. The direct mutagenesis of the residues involved in the disulfide loop eliminates the emetic capacity in SEC1 [215,216].

The major and best-described molecular targets of SAgs are the MHC-II molecules and the TCR. Different regions of the toxins participate in the interaction with these targets. The interaction with the MHC-II is well documented for SAgs of different groups. SAgs mostly interact with the DR isoform of the MHC-II molecule, showing less interaction with DQ and DP. SAgs–MHC-II structures have been already solved, showing the binding surface between the SAg and the DR1 isoform of the MHC-II molecule (Figure 2).

SAgs can interact with the α and/or β chain of the MHC-II molecule, alone or in a complex with a peptide. The interaction with the polymorphic β chain is characterized by strong affinity which is mediated by zinc. SAgs involve highly conserved His and Asp residues, in positions 207 and 209, respectively, to establish coordination with this metal. As Zn^2+^ needs four atoms to complete the interaction, the interaction could involve one more amino acid in position 169, mainly a His or Asp, and the β chain completes the interaction with a His in position 81. SEH, which also displays this kind of interaction with the MHC-II molecule, only contributes with two residues to the union. A water molecule contributes to completing the coordination required by Zn^2+^. All the studied SAgs that contact the MHC-II β chain also contact the bound peptide, which provides more specificity to the binding. On the contrary, the interaction with the invariant α chain usually displays low affinity. SAgs such as SEB and SEC3 that bind to the low-affinity site on the MHC-II α-domain can recognize multiple HLA types (DR, DQ, DP). This is easily explained because the DR α-chain is nonpolymorphic and the SAg makes no contact with the bound peptide [217,218,219].

The interaction with the TCR is very well studied for Group II SAgs. The SE members of this group, SEB, SEC1-3 and SEG are crystallized in complex with the murine Vβ8.2 chain. In addition, the streptococcal SAg, SpeA, which also belongs to this group, is crystallized in complex with this TCR. All the SAgs of this group involve the Asn 25 (SEB numbering) or Asn 24 (SEG numbering) as the hottest spot to interact with the TCR Vβ chain [220]. The SAgs contact the region CDR2, FR2 and 3 and the hypervariable region HV4 of the TCR, except SEG, which does not bind HV4. Despite this fact, SEG shows the highest affinity to the TCR already documented by wild-type SAg. SEG interacts with mVβ8.2 with an affinity in the nanomolar range (500 nM). An analysis of the mVβ8.2-SEG interface elucidates the higher affinity of this complex compared to others as it establishes the highest number of hydrogen bonds in the smallest binding surface [221] (Figure 3). This phenomenon may at least partially explain the early activation of T cells bearing mVβ8.2 by SEG compared to other members of Group II. SEG induces the increase in Vβ8.1+2 T cells in mice 48 h after inoculation, followed by the apoptosis of this cell population. The other members of this group cause this same effect at 96 h, showing a temporary delay compared with SEG biological function. Considering this early immunosuppression, SEG-mVβ8.2 interaction could be advantageous to the pathogen as it may facilitate the colonization of the host [222].

In 2010, Saline et al. solved the crystal structure of the ternary complex between the human MHC class II molecule DR1 loaded with the Influenza hemagglutinin (HA) (306–318) peptide (PKYVKQNTLKLAT), the human TCR (JM22:TRAV27/TRBV19) and the enterotoxin SEH at 2.3 Å of the resolution. The structure reveals that SEH mostly contacts the variable portion of the TCR α chain and only 6% of the variable portion of the β chain. Until now, SEH was the only SE described which presents this kind of binding [223].

With the aim to investigate if other staphylococcal enterotoxins interacting with the DR1 β chain and displaying a similar orientation than SEH over the MHC-II molecule could contact the TCR α chain, we superimposed the structure of SEI-DR1 over the structure of the ternary complex using the DR1 β chain as the template. SEI and SEH show a displacement that puts SEI away from the sphere of the interaction with the variable portion of the TCR α chain (Figure 3). Furthermore, a sequence analysis that took into account all the staphylococcal enterotoxins contacting the DR1 β chain [219,223] shows that none of them conserve the crucial residues to contact the variable portion of the TCR α chain, nor hydrogen bonds or Van der Waals contacts. Considering all the available structural information, SEH would be the only described SE that presents that kind of binding.

The interaction between staphylococcal SAgs and other molecular targets, such as CD1, B7, CD28 and gp130, is well documented. Even though these interactions were not tested in all the staphylococcal SAgs, the conserved residues in their sequence strongly suggest that all of them could bind these molecular targets. Nevertheless, more studies should be conducted to determine the physiological consequences of these bindings. Kaempfer et al. reported in deep detail the interaction between SAgs and B7/CD28 molecules, and as a consequence of this binding, SAgs would improve the contact between B7-2 and CD28, inducing T cell hyperactivation [211,212,213,224].

The interaction between CD1 and staphylococcal SAgs was described by Gregory and collaborators in 2000, and the engagement between SAgs and the CD1a isoform on the monocyte surface disturbs the intracellular calcium flux, altering the biological functions of the target cell [225].

The new molecular targets described increase the cell populations affected by SAgs, implying direct effects outside the immune system.

Latterly, the interaction between gp130, the interleukin 6 receptor and SEA was described by Banke [184]. Gp130 binds SEA with medium affinity inducing phosphorylation of STAT3 in adipocytes. They suggested that an Asp in position 227 would be necessary to stabilize the interaction. Nonetheless, more recently, Noli Truant and collaborators demonstrated that non-classical SAgs lacking Asp227 also interact with gp130 with micromolar affinity [62]. Later, it was shown that SEE can bind human gp130 with a similar affinity to SEA, whereas SEH displays a ten-fold lower affinity [226].

In recent years, Schlievert and collaborators showed that TSST-1, SEB and SEC induce pro-inflammatory chemokine production from human vaginal epithelial cells, being that TSST-1 is much more potent than the others. In addition, using CRISPR-Cas9 knockout CD40 cells, it was shown that this receptor is essential for the chemokine response and that the specificity of TSST-1 for menstrual TSS is in part dependent on the higher-affinity interaction with CD40 than other SAgs, such as SEB and SEC. Furthermore, CD40 seems crucial for the disruption of the human vaginal epithelial barrier by pathogens such as *S. aureus* and might be a potential therapeutic target for drug development [227,228,229].

### 2.5. Actions of SAgs on the Immune System Cells

Conventional immunity mediated by T cells is given by the interaction of an αβ TCR and a complex MHC II-peptide [230,231]. If the TCR recognizes specifically the foreign antigen, the interactions will trigger signaling pathways that will result in the proliferation and differentiation of several clones of T cells [232]. As TCRs are highly diverse molecules, only ~0.01% of the näive T cells will be capable of recognizing a particular antigen [233].

T cell activation by SAgs is distinctive from conventional T cell activation both qualitatively and quantitatively [234]. Given that the characteristic feature of SAgs is their ability to activate T cells in a variable *beta* chain-dependent manner [2], between 5 and 20% of the total T cell population is activated as a consequence of SAg exposure. TCR diversity relies on the CD3 loops due to the V (D) J recombination during T cell development; however, there is a limited number of Vβ possible regions of the TCR, around 50 in humans, and therefore, SAgs can activate many more T cells than conventional antigens. In addition, SAgs can act in the order of picograms per mL [235], inducing a CPA-dependent production of cytokines by T cells characterized by a Th1/Th17 profile [16].

Furthermore, T cell anergy, a phenomenon whereby T cells stop responding to stimuli, has been proposed as an *S. aureus* immune evasion tactic. While the first encounter with classical SAgs induces a rapid clonal expansion of CD4+ and CD8+ T cells with a strong production of Th1/Th17 cytokines, repeated SEA or SEB challenges transduce a hyporesponsive state characterized by T cell deletions and anergy in the remaining SEA/SEB-reactive T cells [236,237]. As a consequence of repeated exposure to SEA, SEB and some streptococcal SAgs, there is an induction of CD25+ FOXP3+ CD4+ T cells Vβ specific with a regulatory profile that expresses IL-10 [238,239] as well as a promotion of CD4+ CD25- T cells [240]. This anergy has been attributed at least in part to CD8+ regulatory suppressive T cells in the case of SEA, SEB, TSST-1 and SEC, and may depend on the concentration of SAgs used to stimulate T cells [241,242,243]. Nonetheless, these regulatory T cells appear to turn ineffective in the massive inflammatory context of TSS [244].

Using SPEA, Sahr and collaborators showed that SAg-stimulated APCs induce pro-inflammatory responses but also promote the initiation of co-inhibitory circuits such as anti-inflammatory cytokines (IL-10), co-inhibitory molecules (PD-L1) and the induction of inhibitory effector programs (IDO) [245]. However, they performed all the assays using only one streptococcal SAg, which seems not enough to generalize for all SAgs.

No reports of T cell anergy using non classical SAgs were found to date; nevertheless, almost every publication using one or two classical staphylococcal SAgs concluded above all SAgs, which may not be true for each of the 30 toxins identified. Although they all are likely to induce anergy, considering that all SAgs stimulate T cells with low specificity, confirmation is required.

Although it is still unclear how exactly anergy occurs, it is clear that not only T cells participate in the reported immunosuppression but also APCs and other immune cells involving several mechanisms, such as anergy, deletion of effector T cells, development of Tregs and induction of tolerogenic profiles in different cell types that should be addressed.

While the superantigenic and enterotoxic effects of SAgs are the most studied mechanisms of the pathogenicity of these toxins, several new works have shown that SAgs have additional capacities.

Dendritic cells (DCs) can uptake SAgs, without inducing maturation, having the ability to maintain themselves as biologically active inside this cell type. Ganem and collaborators reported that DCs effectively incorporated SAgs in a dose-dependent manner by macropinocytosis, retaining unmodified SAgs in endosomes to finally release them to the extracellular medium [214]. Moreover, it was demonstrated that solely the uptake of SEG, SEI, SSA, SEB and SEA did not induce the maturation of DCs [246]. Concerning the retention of biological activity, in this study, SEG conserved its three-dimensional structure since it was recognized by polyclonal antisera during the entire uptake process, inclusion in diverse vesicles and exposure on the membrane, in addition to being able to stimulate T cells in vivo and in vitro. This strongly suggests that SAgs traffic through DCs in intact form. This is not surprising, given that SAgs are known to be very stable and highly resistant to proteases and can resist temperatures of 60 °C or higher, as well as extreme pHs. Earlier studies showed that DCs are more efficient than other APCs, such as B cells or monocytes, to initiate T cell proliferation by presenting TSST-1, SEA, SEB and SEE. Results suggested that picomolar levels of these SAgs are required on DCs to maximally stimulate T cells. This would be related to the high amount of MHC II expressed on DC surface [247].

An ex vivo study in mice showed that DC maturation and migration induced by SEB and TSST-1 require T cell activation. Furthermore, it was evidenced that SEB is also able to upregulate MHC II, CD40, CD205 and CD86 markers [248]. Human DC capability to upregulate MHC II, CD86, CD80, CD83 and CD54 in the presence of SEB was also demonstrated. In contrast with previous data, Coutant et al. found no effect on CD40 expression. An increase in the production of TNF-α by human DCs induced by SEB was shown, compared to other compounds such as LPS. Enhancement of IL-12p270 human DC production by SEB remains controversial because some suggest that it occurs, and others affirm that it does not affect its production [249,250].

As mentioned before, SAgs interact simultaneously with TCR on T cells and MHC II on APC and they have to be in close contact, which occurs in the lymphoid tissues. It was shown in mice DCs that they can internalize SEG, SEI and SSA as biologically active molecules and recycle them into the cell membrane. This fact does not cause DC maturation. SAgs were found intact in the acidic cells compartment and remained active. These results suggest that intracellular trafficking of SAgs in DCs improves their local concentration and promotes their encounter with T cells in lymph tissues [214].

SAgs induce cytokine production in APCs, which promote Th1/Th17 profiles [16]. However, an in vitro study in human DCs suggested that SEB may induce T cell immunoglobulin mucin domain (TIM) 4 expression in these cells when it is processed via TLR2 and NOD1 pathways. The interaction between TIM4 in APCs and TIM1 in CD4+ naïve T cells induces a Th2 profile [250]. These results agree with the idea that SAgs may have diverse effects on immune system cells depending on the interaction pathway.

Plasmacytoid dendritic cells (pDCs) are another innate cell subpopulation that may interact with SAgs. An assay in mice demonstrated that SEA increases the number of pDCs in lymphoid organs and promotes the expression of CD86 and CD40. Furthermore, pDCs would strictly require the presence of IFN-γ, in addition to the interaction with T cells, for maturation. In contrast, DCs do not depend on this cytokine for its activation by SAgs [251].

It has been reported that the interaction of SEB with monocytes enriched from PBMCs induced cell death and IFN-γ production [252]; however, the presence of the remaining T cells, a possible source of IFN-γ, cannot be discarded in those studies. Furthermore, *egc* SAgs inhibited monocytic proliferation in a dose-dependent manner, promoting cell death by apoptosis, and to a lesser extent, by necrosis. The significant production of TNF-α, IL-6 and IL-12 but not of IFN-γ was detected in this scenario [62]. Only SEI induced early apoptosis at the times assayed, similar to SEB that would induce TNF-α dependent apoptosis in THP-1 cells (a cytokine that the classical SAg induces to a greater extent than *egc* SAgs [253]), suggesting that other intracellular pathways might be involved in the death processes promoted by different SAgs. In concordance, Ulett and Anderson [254] proposed that death pathways could vary not only between different cell types but considering the toxin and the molecular conditions of the stimuli. Furthermore, the absence of CD4+ T cells in the cultures of THP-1 may explain the lack of IFN-γ production [255].

Macrophagic cells obtained by PMA-differentiated THP-1 cells inhibited their proliferation and produced pro-inflammatory cytokines in response to *egc* SAgs. In addition, SEI induced significant cell death by apoptosis in this cell type, while SEG, SEM and SEO induced death by apoptosis as well as by necrosis. SEB is the only classical SAg reported to induce apoptosis in THP-1 cells PMA-differentiated to macrophages by caspases 3 and 8 [252], but this does not necessarily explain the death mechanism of *egc* SAgs. All these results not only demonstrate that damage mechanisms are variable according to each toxin, between other factors, but that the response triggered by different SAgs simultaneously may be more diverse than expected.

Despite visible differences, several studies have shown evidence that immune activation generated by classical and new SAgs is similar [256], inducing an intense profile of the Th1/Th17 immune response [235,257,258,259,260].

Although Dauwalder (2006) suggested that *egc* SAgs would have a different inflammatory potential than classical SAgs, which may explain the different severity of the symptoms caused by *S. aureus* carriers of one or another type of SAg, this has not been confirmed by other authors. Possibly, their results could be attributed to differences in the production or the purification of SAgs, considering that they used a purchased SEA and a lab-produced SEG, or even to the concentrations used. The most accepted hypothesis is that the differences between the effects of classical and *egc* SAgs lie in a complex network of regulatory pathways that determine the moment and the conditions in which *S. aureus* produces them [256,261], resulting in much lower concentrations of *egc* SAgs compared to classical SAgs [87,88,89], as was mentioned in Section 2.2.

In contrast with αβ T cells, γδ T cells constitute between 1 and 5% of PBMCs and are mainly concentrated in skin and mucosa. Eighty percent of γδ T cells in peripheral blood co-express the chains Vγ9 (alternatively Vγ2) or Vδ2, and it has been demonstrated that when activated, they can act as antigen-presenting cells [262], in addition to having the cytotoxic capacity and a great source of cytokine production. While some works stated that some classic SAgs, such as SEA, SEB and TSST-1, can stimulate directly or indirectly different subsets of γδ T cells, differences are referring to each SAg [260,263,264]. These suggest that the activation of this specific cell type could be related to a characteristic affinity of each subset of γδ TCR and its capacity to interact with one or another SAg, indicating that the residues involved in the binding with the αβ and γδ TCR are not shared. The capacity to stimulate this subpopulation of T cells has been extended to other SAgs; however, no evidence supports this fact for non-classical SAgs.

Furthermore, it has been reported that SEB activates NKT cells, and that SEA and SEB can interact with invariant NKT cells (iNKT) [265,266,267], inducing a pathogenic role of this cell type in SST. Moreover, classical SAgs interact with B cells [268,269,270], and IgE antibodies against superantigens have been detected [271,272].

It is demonstrated as well that SEA, SEB, SEE and TSST-1 can interact with mast cells [60,273,274,275], and the novelty that SEB activates mucosal-associated invariant T cells (MAIT cells) to produce high levels of INF-y, TNF-α and IL-2 and then induces anergy in this cell type [276]. MAIT cells are an unconventional T cell subset with a semi-invariant T cell receptor that recognizes antigens presented in the context of classical MHC-like molecules [277] and are involved in microbial immunity displaying protective and pathogenic responses [278,279]. Again, there is no information about the effect of non-classical Sags on these cells.

SSL toxins are another group of exotoxins of *S. aureus* that are unable to induce T cell proliferation but can contribute to immune evasion [48], interfering with complement and neutrophil function [280]. Similarly, Se*l*X can bind both monocytes and neutrophils, impairing neutrophil activation [281,282], and is considered the first line of defense against *S. aureus* [283].

Although the activation of other cell types could have a considerable impact on the immune system response, almost every study published has been developed using the less prevalent classical SAgs, and it is not clear if their effects can be extended to the newer SAgs, which are widely distributed, such as the *egc* operon SAgs.

Due to their ability to interact with several components of the immune system, SAgs have been clinically used as immune modifiers in neoplasm treatment and other pathologies.

### 2.6. Superantigens as Therapeutic Tools

Considering the complexity of oncological processes, altered superantigens with reducing systemic toxicity that conserves the ability to eliminate tumoral cells have being considered as alternative therapies. Indeed, several cancer cell lines are effectively attacked by PBMCs activated by classical SAgs. Furthermore, various SAgs mutated in the MHC II contact region have been created willing to keep the antitumoral effects of SAgs, thus reducing systemic toxicity. Considering this strategy to diminish side effects, SAgs continue being extensively investigated for oncological applications either alone or in combination with classical anticancer drugs [284]. It was described that PBMCs stimulated with SEA display the ability to promote the death of human lung carcinoma A549 cells [285]. Similar results were observed when SEB-stimulated PBMCs induced apoptosis on transitional cell carcinoma cells [286]. These first studies were carried out with wild-type superantigens, which conserved intact their potent inflammatory activity. With the aim to avoid SAg systemic toxicity, various SAgs altered in hot spot regions have been created. That is the case for SEA D227A, which was conjugated with an antitumor Fab antibody; the fusion protein conserved the capacity to activate T cells but reduced the binding to the MHC-II molecule 500 times, compared to wild-type SEA [287]. Moreover, SEC2 double mutant, T20L/G22E, inhibited the growth of S180 sarcoma with less toxicity in mice [288].

Other alternatives to reduce systemic toxicity in using SAg as therapeutic tools include mutant SAgs that were designed as a part of chimeric single chain antibodies (scFv) specific to tumoral antigens. These SAgs have been mutated in the binding site to the MHC-II molecule with the aim of reducing their superantigenicity. These mutations reduced their affinity to the MHC-II molecule, and the cytotoxic effect on MHC class II- expressing cells [289]. Recently, a new generation of antibody–superantigen fusion proteins was designed, in which the SAg, in this case SEA, was split into two fragments. Individually, each fragment remains inactive, but when the biological formulation reaches the target cell by binding to cell surface antigens, it dimerizes and the SAg regains its biological functionality activating T cells. The effective split SEA design would not affect MHC class II-expressing cells, but when bound to a tumor antigen via a targeting moiety it would activate a T cell response [290].

Non-classical SAgs were also investigated in the immunotherapy field. The *egc* operon superantigens SEG and SEI were linked to the endogenous human MHCII HLA-DQ8 allele in humanized mice inducing a potent antitumor response and extending life in an established melanoma mice model [291].

In addition, the combination of mutant SAgs with other molecules to create chimeras that specifically stimulate the immune system with low toxicity is another strategy not fully studied yet, but with promising results, as in the case of mutant SEG used as a candidate for Chagas vaccination [292]. In that work, a chimeric molecule comprising a *T. cruzi* antigen and a non-toxic form of SEG was used as a novel immunogen to confer protection against *T. cruzi* infection. The mutant SAg used retained its ability to trigger classical activation of macrophages without affecting T cells, because this Th1 profile was adequate to eradicate intracellular protozoa such as *T. cruzi*, proving to be an effective immune modulator against this parasite.

These uses of SAgs as therapeutic agents that permit taking advantage of these toxins allow us to compare them to Dr. Jekyll, but it should not be forgotten that SAgs usually also act as Mr. Hyde, causing several affections and worsening disease conditions.

Unfortunately, everything in nature has its Dr. Jekyll and Mr. Hyde side, and superantigens are one of the most crucial toxin threats in warfare or bioterrorism as they are resistant to heat and can be administrated in contaminated water or as aerosols in the air.

It is our choice to use these toxins either like Dr. Jekyll or Mr. Hyde.

## 3. Concluding Remarks

Microorganisms evolve to evade the host immune system. All mammal pathogens have to face a strong and complex immune response, designed to fight them back. Manipulating this kind of response could be the most effective mechanism to spread the infection. This situation is the ideal scenario to develop a chronic infection and reach an equilibrium between the effector mechanisms of the immune system and the war for survival of the pathogen.

*Staphylococcus aureus* causes an acute infection where quick spread and colonization are essential to reach its goal. Considering that inflammation is crucial for the resolution of most bacterial infections, it is not clear how a pro-inflammatory state may help bacteria succeed. However, the inflammation induced by SAgs seems to impair activation and recruitment of important effector cells, promoting a temporal host immune suppression and the survival of *S. aureus*. In view of the numerous diseases in which SAgs participate or are implicated, it can be ascertained that Mr. Hyde’s side of SAgs is the easily visible face of these toxins.

For many years, it was considered that SAgs only affected T cell function, driving them to anergy or apoptosis, and promoting host immune suppression. As described throughout this work, SAgs would not only exert their function on T lymphocytes in the presence of antigen-presenting cells, but they would also be capable of eliciting different effects on monocyte/macrophage effector cells, ɣδ T lymphocytes, lymphocytes B, neutrophils and other cells not belonging to the immune system, such as adipocytes. In general, this action would lead to nonspecific pro-inflammatory conditions that would deplete effector cell populations, thus being inhibited to eradicate the bacteria that produce them, promoting a state of host immunosuppression from the first to the last stages of the infection, which would favor bacterial propagation. However, this intricate scenario, which is gradually being cleared, can be reversed, by taking advantage of the action of SAgs while restricting their effects for human benefit, thus revealing the protective yet still unclear Dr Jekyll side of SAgs.

## Figures and Tables

**Figure 1 toxins-14-00800-f001:**
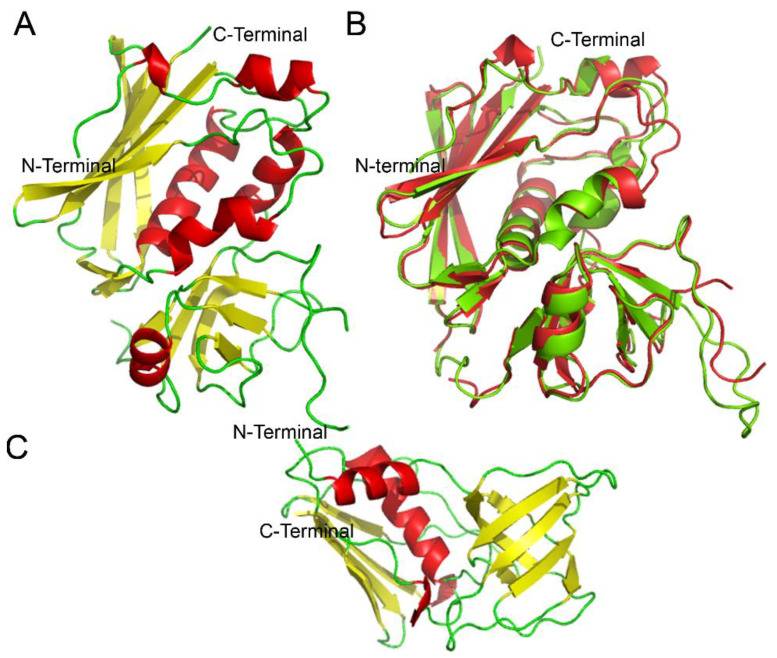
Structural features of Staphylococcal superantigens. (**A**) Overall structure of staphylococcal enterotoxin G (SEG) PDB accession number 1XXG. The general structure of SEG is displayed as a cartoon, and the secondary structures are colored yellow, β strands; red, α helixes; and flexible loops, green. The N-terminal domain and the C-terminal extreme are both located in the same domain of the molecule. (**B**) Superimposition of SEG and SEB. SEG overall structure (red) was superimposed over SEB structure (green) with an RMS of 0.714 Å, suggesting high similarity. (**C**) Overall structure of TSST-1, showing a simpler general structure than that of other staphylococcal enterotoxins. All the figures were performed using the PyMOL software.

**Figure 2 toxins-14-00800-f002:**
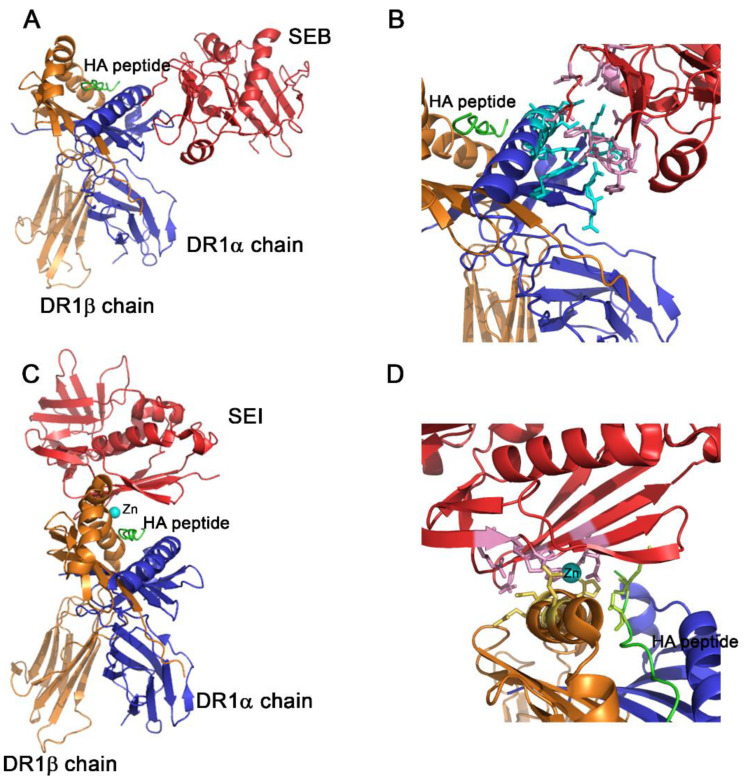
Structure of the SAg-HA-HLA DR1 complex. (**A**) Ribbon diagram of SEB-HA-HLA DR1 complex. (**B**) The interface of the interaction is shown in detail. SEB compromises residues in positions 43, 44, 45, 46, 47, 67, 89, 92, 94, 95, 115 and 209. HLA DR1α chain, involves the residues: 13, 17, 18, 36, 37, 39, 57, 60, 61, 63, 67 and 68. Non-contacts are found between the peptide and SEB or SEB with the DR1β chain. (**C**) Ribbon diagram of the SEI-HA-HLA DR1 complex. (**D**) The interface of the interaction is shown in detail. The interaction is coordinated by Zn^2+^. This metal ion interacts with His81 of the DR1β chain and His169, His207 and Asp209 of SEI. SEI compromises residues in positions 98, 100, 105 and 211 to contact the residues 307 and 309 of the hemagglutinin (HA). No contacts are found between SEI and the DR1α chain. In all panels, the superantigen is colored red; the HLAD1α chain, blue; and the DR1β chain, orange. Zn^2+^ is represented as a sphere in cyan and the HA peptide, green. The residues conforming the interaction surface are represented as balls and sticks and colored pink (SEB or SEI); cyan, HLAD1α chain; yellow, DR1β chain; and light green, HA peptide. The figures were performed using PyMOL and the analysis of the structures was carried out using the CCP4i suite program.

**Figure 3 toxins-14-00800-f003:**
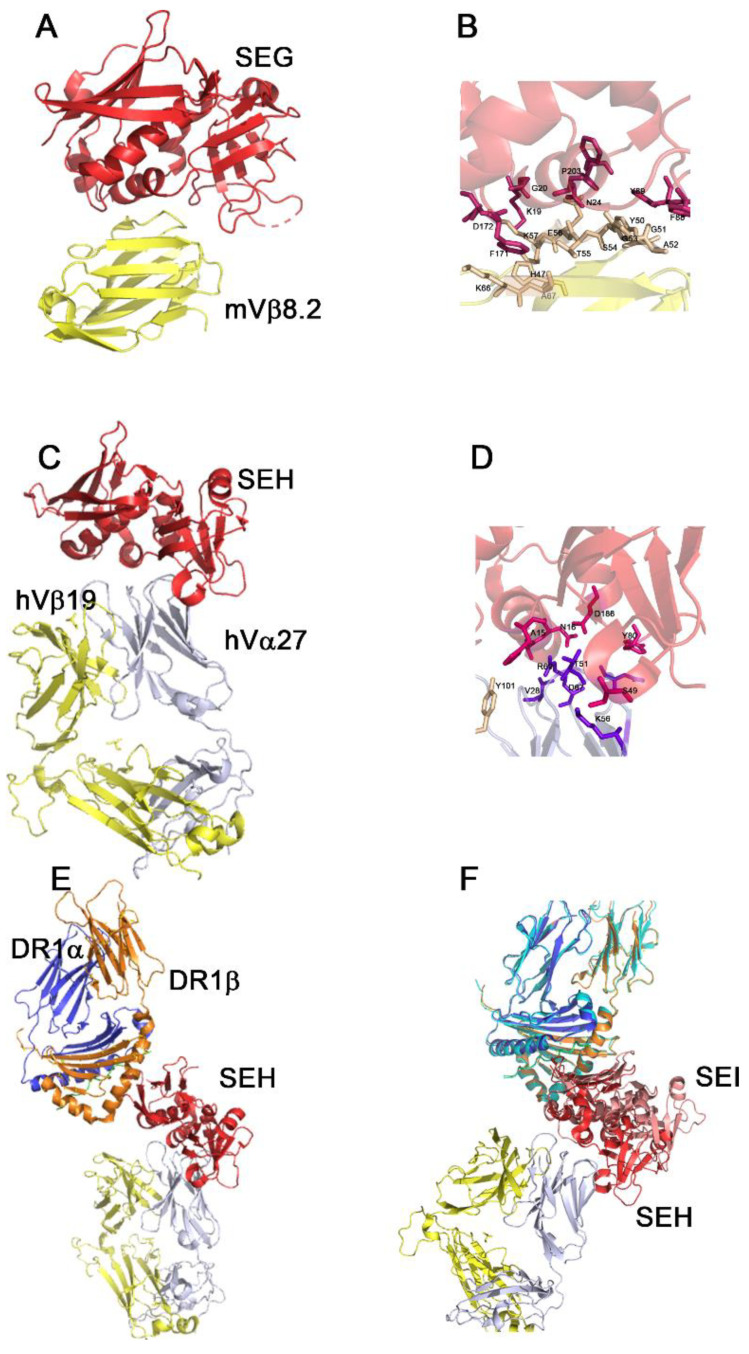
Crystallographic structures of the SAg-TCR interaction. (**A**) Ribbon diagram of SEG-mVβ8.2 complex. SEG is colored red and the TCR β chain, yellow (**B**) as shown in detail. SEG residues are colored pink and mVβ8.2 residues, wheat. Residues are indicated with a one letter code and numbered. (**C**) Ribbon diagram of the SEH–human TCR complex. SEH is colored red and the TCR, light blue (α chain) and yellow (β chain). (**D**) The interface of the interaction is shown in detail. SEH residues are colored hot pink, hVα27 chain residues are colored violet and hVβ19 residues are colored wheat. Residues are indicated with a one letter code and numbered. (**E**) Ribbon diagram of the SEH-TCR-MHC-II tri molecular complex. SEH is colored red, HLA-HA-DR1 is colored blue (α chain) and orange (β chain) and the TCR as indicated in C. (**F**) Superimposition of the SEI-HA-HLADR1 complex over the trimolecular complex using the DR1 as template. SEI shown in pink is clearly away from the interaction surface with the TCR α chain. The figures were performed using PyMOL and the analyses of the structures were carried out using the CCP4i suite program.

**Table 1 toxins-14-00800-t001:** Staphylococcal SAgs.

SAg	Type	Year of Discovery	Phylogenetic Group
SEA	Classical	1962	III
SEB	Classical	1962	II
SEC	Classical	1965	II
SED	Classical	1967	III
SEE	Classical	1971	III
TSST-1 (SEF)	Classical	1981	II
SEG	New	1998	I
SEH	New	1994	III
SEI	New	1998	V
SE*l*J	New	1998	III
SEK	New	2001	V
SEL	New	2001	V
SEM	New	2001	V
SEN	New	2001	III
SEO	New	2001	III
SEP	New	2005	III
SEQ	New	2002	V
SER	New	2004	II
SES	New	2008	III
SET	New	2008	I
SE*l*U	New	2003	II
SE*l*V	New	2006	V
SE*l*W	New	2012	III
SE*l*X	New	2011	I
SE*l*Y	New	2015	I
SE*l*Z	New	2015	I
SE*l*26	New	2018	V
SE*l*27	New	2018	II
SE01	New	2017	III
SE02	New	2020	II

## Data Availability

Not applicable.

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
