# Peer review of "Superantigens, a Paradox of the Immune Response"

_toxins, 2022, doi:10.3390/toxins14110800_

Round 1
Reviewer 1 Report
This review describes the family of superantigens found in Staphylococcus aureus. Detection of sag genes and proteins are discussed followed by an overview of Sag-associated human diseases, Sag classification, protein structure and interaction with immune receptors. The potential use of Sags in anti-cancer therapies is briefly discussed.
My main concern about this review is the lack of novelty and the lack of critical analysis. There have been numerous reviews on this topic over the last 2-3 decades and this review doesn’t add much to the overall story. It is also quite superficial. Over 250 articles are cited but the author(s) skim trough them without much critical evaluation of the topic. Why are superantigens considered a paradox of the immune response (title)? The ‘Dr. Jekyll and Mr. Hyde theme’ is discussed in one short paragraph (11 lines) at the end.
The manuscript also needs careful proof-reading for English language.
Specific points:
Line 41: Mycoplasmas are not Gram-negative. They are related to Bacilli (Gram-positive) but lack a cell wall and therefore don’t stain.
Line 55: Header missing
Line 70: Toxic Shock Syndrome Toxin
Table 1; please add references
Line 165: what is the detection limit?
Line 189: how is oxygen introduced?
Line 337: sag genes were found in blood samples?
Line 382: what is the link between globular structure and pathologies? What is meant by ‘remaining active … through the lysosome”? Sags work outside the cell.
Line 389: the table shows sequence homologies, not structural homologies
Line 396: TSST has low sequence homology with other staphylococcal Sags, but still a conserved structure (also line 413)
Line 409: what is meant by ‘extreme’?
Line 434: should that be HLA-DR1?
Line 441-447: these positions only refer to SEI. Position 169 is also involved (not could be). The His81 is from the HLA-DR beta chain.
Line 448: how does a water molecule complete the interaction?
Line 453: should that read “the HLA alpha domain …. can recognise multiple HLA types (DR, DQ, DP)”? The Zinc binders also recognise multiple DR alleles as they all carry the conserved His residue.
Line 464: I believe the affinity is higher in the trimeric complex
Line 467: what is meant by “early” activation? Why would it confer a faster rate of colonisation?
Line 537” variable beta chain
Line 542: should that be pMol? A weight unit doesn’t make much sense here.
Line 563: there is no indication that this might not be true for other Sags as they all stimulate T cells with low specificity.
Line 572: more info is required on how Sags that are taken up by dendritic cells maintain their function (binding to TcR).
Line 622: confusing sentence
Line 638: this needs clarification
Line 664: similar how? TSST has Sag activity, but not the SSLs
Line 677: ‘numerous’ seems exaggerated. What was the purpose of the mutations in the MHC II binding site?
Line 684: how was SEG used as a ‘candidate for Chagas vaccination’?
Author Response
We appreciate the caring and valuable revision that Reviewer 1 did to this manuscript, which greatly improved its clarity and quality. We are positive in answering all the concerns of Reviewer1. We provided a file with the answers point by point and the lines where the changes were introduced. (PLEASE SEE THE ATTACHMENT)

Reviewer 2 Report
This is a generally well-organized and well-written manuscript. The major immune mechanisms of superantigen alteration of the host are provided. However, there are some changes that need to be addressed:
1. The authors early-on discuss enterotoxin F, as if it were an enterotoxin. SEF was changed to TSST-1 because there is no emetic activity associated with it. Thus, TSST-1 does not meet the criteria of being an enterotoxin. This is later addressed correctly, but upon first discussion, the reason sould be give. Additionally, those in the field know that Dr. Begdoll's SEF was contaminated by SEA, a known enterotoxin. That was accounting for the misleading emetic activity.
2. The authors address the factors that affect TSST-1 production and list temperature, carbon dioxide etc as important. However, by far the most important factor, as discussed later is oxygen. That is the reason for the tampon association. The authors also do not seem aware that there is a background TSS. Schlievert published that this background level explains the apparent but not real association with menstrual cups. it is oxygen trapped in tampons that explains their association.
3. A minor point (line 274) please change "endotoxins' to a different term. All superantigens amplify by one-million fold the host susceptibility to real endotoxin from Gram-negative bacteria. This point also should be mentioned as it likely really explains why people die of TSS.
4. The authors imply the Dr. Jekyll Mr. Hyde aspects of superantigens. They also imply that the production of superantigens is not well-understood. However, Stach et al. have shown that the EGC superantigens are produced at much lower concentrations than for example TSST-1 and SEsB and C, the three major causes of TSS. This makes the EGC more likely to be important for colonization of S. aureus than overt disease production.
5. Through CRISPR cas9 technology, the Schlievert group has shown that CD40 is likely to be the first immune cell receptor for superantigens as present on Non-standard immune epithelial cells. This should be addressed
Author Response
We appreciate the caring and valuable revision that Reviewer 2 did to this manuscript, which greatly improved its clarity and quality. We are positive in answering all the concerns of Reviewer 2. We provided a file with the answers point by point and the lines where the changes were introduced. (PLEASE SEE THE ATTACHMENT)

Round 2
Reviewer 1 Report
The authors have addressed the specific issues and provided an improved version of the manuscript. However, there are still some issues remaining. Most of all, despite proof-reading by a professional translator, there are still many awkward sentences and phrases. I would strongly recommend to have the manuscript proof-read by a colleague.
The “Dr Jekyll” section has been expanded but given the title I believe it should be still further expanded. What are the antitumor effects of SAgs? How were they mutated? How exactly do the Sag-Ig chimera work? Why are they targeted to tumor cells which I assume don’t carry MHCII (or are they lymphomas?). How does the T. cruzi antigen-Sag chimera work? Is it to deliver the antigen to an APC? I would also recommend to introduce a subheader for this section.
Specific points:
- Line 42: references required
- Line 50: write out SpeA. Remove “//”
- Line 52: ‘codification’ is an unusual term.
- Line 56: references required
- Line 73: what properties?
- Line 80: codifying is an unusual word (suggest encoding)
- Line 82: awkward sentence
- Line 86: what is meant by ‘complex form’?
- Table 1: the phylogenetic groups are only explained much later in the text. I think the flow of the manuscript could be improved, e.g by moving the disease section closer to the end.
- Line 104: taxonomical names in italics
- Line 110: weird phrase
- Line 111 (and throughout manuscript): protein names starting with capital letter, gene names in small letters and italics. This includes families (sag gene; SAg protein).
- Line 112: what does percentage refer to?
- Line 122: meaning unclear
- Line 130: these are some very old articles. Are there any newer studies using more reliable tests such as ELISA?
- Line 147: components?
- Line 148: replace ‘proven’ with ‘validated’
- Line 151: replace ‘but’ with ‘and’
- Line 152-155: the amount alone might not be that relevant as the potency differs between SAgs
- Line 159: awkward sentence
- Line 168: awkward sentence
- Line 197: confusing sentence
- Line 204: what is meant by ‘increment’?
- Line 250: I don’t think pneumonia is only caused by CA-MRSA strains
- Line 253/274: ‘isolates’ instead of ‘isolations’
- Line 279: ‘foodstuff’ is an uncommon term in science
- Line 293: not sure why TSST is mentioned here as it is not emetic
- Line 294: amino acid instead of aminoacidic
- Line 298” what is meant by ‘jejune tonic’
- Line 319: do SAg generate autoimmune cells or reactivate existing cells that are controlled by peripheral tolerance?
- Line 321: the autoimmune theory in KD is controversial. Have these references actually shown autoimmune cells?
- Line 335: how do endotoxins fit in here? Synergistic effects?
- Line 356: cross-reactive T cells, not SAgs
- Line 360: confusing. Does it refer to S. aureus isolates carrying the seh gene?
- Line 364: Suggest “antibodies against SEB and TSST”
- Line 367: as mentioned earlier (line 319)
- Line 378: infiltration of what?
- Line 383: how do SAgs activate B cells? Or does it refer to B cell superantigens which are completely different (non-related) proteins?
- Line 385: weird phrase
- Line 405: what is meant by ‘lowest level’?
- Line 411: references required for this new paragraph
- Line 417: what sequences (nct or aa)?
- Line 424-429: lack of flow. What is meant by ‘toxins have more structural complexity’?
- Line 461: Zn should be Zn2+
- line 474/477: These positions differ between individual SAgs
- line 488: m = murine
- line 490: TcRVbeta chain (not TcR)
- line 497-502: can a lower binding affinity alone really explain a delay in T cell stimulation seen over several days?
- Line 503: how would this explain a faster colonization rate?
- Line 591: weird sentence
- Line 600: only IL-10?
- Line 622: what is meant by ‘retained enough structure’?
- Line 649: what is TIM4? Please explain
- Line 662: weird phrase
- Line 691: what is the range for those two groups?
- Line 709: confusing sentence
- Line 719: why mention SSLs here? They are not SAgs.
Author Response
We want to thank again for the rigorous revision that Reviewer 1 did of this work. We appreciate all the observations which improve the manuscript a lot. All the changes were introduced in blue. We provided a file with the answers
Sincerely
